# Impact of COVID-19 on the Health of the General and More Vulnerable Population and Its Determinants: Health Care and Social Survey–ESSOC, Study Protocol

**DOI:** 10.3390/ijerph18158120

**Published:** 2021-07-31

**Authors:** Carmen Sánchez-Cantalejo, María del Mar Rueda, Marc Saez, Iria Enrique, Ramón Ferri, Miguel de La Fuente, Román Villegas, Luis Castro, Maria Antònia Barceló, Antonio Daponte-Codina, Nicola Lorusso, Andrés Cabrera-León

**Affiliations:** 1Andalusian School of Public Health (EASP, Escuela Andaluza de Salud Pública), 18080 Granada, Spain; carmen.sanchezcantalejo.easp@juntadeandalucia.es (C.S.-C.); antonio.daponte.easp@juntadeandalucia.es (A.D.-C.); 2Institute of Biosanitary Research, ibs.Granada. (IBS-E-10), 18080 Granada, Spain; 3Department of Statistics and Operations Research, University of Granada, 18014 Granada, Spain; mrueda@ugr.es (M.d.M.R.); rferri@ugr.es (R.F.); luiscastro193@gmail.com (L.C.); 4Research Group on Statistics, Econometrics and Health (GRECS), University of Girona, 17003 Girona, Spain; marc.saez@udg.edu (M.S.); antonia.barcelo@udg.edu (M.A.B.); 5Network Biomedical Research Center of Epidemiology and Public Health (CIBERESP), 28029 Madrid, Spain; 6Andalusian Institute of Statistics and Cartography, 41071 Seville, Spain; iria.enrique@juntadeandalucia.es; 7Demométrica, Market Research and Public Opinion, 28001 Madrid, Spain; mfuente@demometrica.com; 8Andalusian Health System, 41001 Seville, Spain; romanp.villegas@juntadeandalucia.es; 9Andalusian Health and Environment Observatory (OSMAN), Andalusian School of Public Health (EASP), 18080 Granada, Spain; 10Health Surveillance Service, Department of Health and Families, Andalusian Regional Government, 41020 Seville, Spain; nicola.lorusso.sspa@juntadeandalucia.es

**Keywords:** public health, health determinants, health inequalities, COVID-19, SARS-CoV-2, vulnerable populations, real-world data, surveys, population registries, machine learning

## Abstract

This manuscript describes the rationale and protocol of a real-world data (RWD) study entitled Health Care and Social Survey (ESSOC, Encuesta Sanitaria y Social). The study’s objective is to determine the magnitude, characteristics, and evolution of the COVID-19 impact on overall health as well as the socioeconomic, psychosocial, behavioural, occupational, environmental, and clinical determinants of both the general and more vulnerable population. The study integrates observational data collected through a survey using a probabilistic, overlapping panel design, and data from clinical, epidemiological, demographic, and environmental registries. The data will be analysed using advanced statistical, sampling, and machine learning techniques. The study is based on several measurements obtained from three random samples of the Andalusian (Spain) population: general population aged 16 years and over, residents in disadvantaged areas, and people over the age of 55. Given the current characteristics of this pandemic and its future repercussions, this project will generate relevant information on a regular basis, commencing from the beginning of the State of Alarm. It will also establish institutional alliances of great social value, explore and apply powerful and novel methodologies, and produce large, integrated, high-quality and open-access databases. The information described here will be vital for health systems in order to design tailor-made interventions aimed at improving the health care, health, and quality of life of the populations most affected by the COVID-19 pandemic.

## 1. Introduction

### 1.1. Background

Currently underway are a large number of studies investigating the evolution of the 2019 coronavirus disease (COVID-19) and the impact it is having in terms of the number of infected patients, hospital admissions, and deaths [1,2,3,4,5,6], as well as on the mental health and well-being of the population [7]. Nevertheless, very few reports providing information concerning clinical, contextual and citizens’ general perceptions pertaining to the pandemic’s impact and evolution since its onset are being prepared [8,9,10].

The Spanish Government officially declared a State of Alarm on 14 March 2020 (Spanish Royal Decree 463/2020 [11]) in the face of the global public health emergency caused by COVID-19. Among other actions, it ordered individuals’ freedom of movement to be limited (Article 7, “Limitación de la libertad de circulación de las personas”), which was subsequently restricted even further through other decrees (hereinafter referred to as confinement). These limitations have led to a series of still little-studied problems in the population. In this sense, recent reviews of epidemic outbreaks and subsequent confinements have concluded that these actions have very negative and long-term impacts on mental health [12,13]. Likewise, this current pandemic is also seeing such negative effects, albeit in the short-term [14,15,16,17]. Furthermore, all studies agree on the urgent need for more evidence regarding the kind of impacts being experienced, in particular, evidence gathered from among the most exposed populations as well as those populations in a situation of greater vulnerability such as minors in foster care, ethnic minorities, the elderly or people with chronic diseases [8,18,19,20,21].

COVID-19impact on ethnic minorities

The results of different studies on the impact of COVID-19 on the migrant population and ethnic minorities have revealed a higher incidence of the disease in these population groups in relation to deficient conditions in terms of socioeconomic deprivation, comorbidities, unemployment, and liveability [22,23,24]. As in the case of the Roma population, this situation may be aggravated in the migrant population as they have less access to health care because of the deterioration of their socioeconomic and administrative conditions or their difficulty in understanding the prevention and health care information provided due to language limitations [25,26].

COVID-19impact on people with chronic diseases

The results of some studies into the effects of pandemics on the health of people with chronic diseases have highlighted the need to adapt health care to the confinement situation by introducing measures such as remote consultations [27,28]. Furthermore, these studies revealed that throughout the confinement period, the symptoms of some diseases, such as Alzheimer’s [29], diabetes [30] or chronic obstructive pulmonary disease (COPD) [31], were exacerbated. It is well known that health care can be affected during health emergencies and pandemics as a result of resources being diverted towards more urgent areas [32,33,34].

COVID-19impact on elderly population

In Spain, 38% of confirmed COVID-19 cases, almost half of the hospitalisations due to the virus, and more than 85% of the deaths, are among people over 70 years of age [35,36] Internationally, it has been confirmed that the elderly are at greater risk of hospitalisation and developing severe symptoms [37,38], while people over 60 and those with underlying conditions such as hypertension, diabetes, cardiovascular diseases, chronic respiratory diseases, or immunosuppression, are at greater risk of severe infection and death [39,40]. Furthermore, these extreme pandemic-related incidence and mortality rates among the elderly population are further aggravated by its psychosocial impact, with a higher risk of depressive feelings [41,42,43].

The aforementioned is a clear reflection of the tragic COVID-19 situation that is currently being experienced, and may be even harsher in deprived socioeconomic contexts [9], although these results must be further explored and there is no evidence, in this respect, in our context. Furthermore, the situation caused by the pandemic means that coordination between social and health care services, together with the provision of resources, are even more necessary and urgent than ever [44].

With respect to the health impact of COVID-19 in Spain, several studies have been launched such as the Spanish Seroepidemiology Study (Estudio Nacional de Seroepidemiología, Spain) which is based on the data obtained through a probability population-based macro-survey that provides estimates down to the provincial level, monitoring the evolution of the epidemic, and information obtained from patients’ medical records [36,45,46]. Most of the studies found are using cross-sectional designs, non-probabilistic samples, and web surveys [47,48] and represent an agile and simple alternative for collecting a large amount of data quickly and offering results practically in real-time; something which is crucial to responding to situations such as the one currently being experienced. However, they do have a number of disadvantages and limitations. For instance, they do not allow for the impact on and evolution in certain populations in more vulnerable positions to be determined, nor do they allow valid estimates to be obtained which would, in turn, allow the level of error to be limited to the appropriate standards to be conclusive. The reason for this is because they are not usually processed with a view to solving problems resulting from a lack of coverage and response, or from selection bias in the sample [49].

### 1.2. The ESSOC Study Framework

The Health Care and Social Survey (ESSOC, Encuesta Sanitaria y Social) research project arose from the need to provide data on the impact of COVID-19. Its results can then be considered when making decisions to prepare and provide an effective Public Health response in the different populations affected, especially the most vulnerable such as, among others, the elderly, the chronically ill, or those at risk of exclusion. ESSOC focuses on Public Health [50] and relies on the participation of society to obtain information on people’s health and their quality of life in order to be able to intervene both individually and collectively in the face of the pandemic (Table 1). The research project is based on a real-world data [51] design that integrates observational data extracted from multiple sources of differing natures and perspectives, i.e., both probabilistic and administrative. Thus, it will generate a very large amount of linked data from differing sources based on longitudinal probabilistic population surveys (more than 22,000 interviews with information on more than 700 variables over three years on the economic situation, state of health and well-being from the perspective of the citizens involved), and sources based on clinical (more than one million records between COVID and non-COVID cases with information on more than 110 variables over two years), epidemiological (almost 600,000 cases with information on more than 25 variables over one year), demographic (from the 8.5 million people residing in Andalusia), and environmental records (more than a million records with information on more than 20 variables over 10 years–i.e., data from an administrative perspective).

In addition, the data analyses will be performed using advanced statistical, sampling and machine learning techniques, which will allow for new research methods to be developed and implemented [52]. The research follows an Open Science approach [53] in terms of disseminating its results, methodologies, processes, and collected data, which will be distributed, reused, and freely and openly accessible to not only the scientific, academic, clinical, and public health managerial community, but also to society at large and, in particular, those population groups identified as being at greater risk of vulnerability to COVID-19. Its management model is that of collaborative and multidisciplinary research, facilitating the creation of a context of scientific cooperation in the fields of public health, health care, public administration, data science, environmental and demographic sciences, and social sciences (Table 1). Finally, the study region, Andalusia, is the most populated (8.5 million inhabitants) and the second largest in area of the 19 regions in Spain. It is also a region of interest within Europe as it is the fifth most highly-populated region. The ESSOC research project is included in the Oxford Supertracker global directory of policy trackers and surveys related to COVID-19 [1].

### 1.3. Hypotheses

Perceptions of general health, mental health, and emotional well-being could have deteriorated in the short- (one year) and mid-term (three years) since the beginning of the pandemic, with a greater impact perhaps being observed in women, young people, and those diagnosed with COVID-19.The socioeconomic, psychosocial, behavioural, occupational, environmental, and clinical determinants of health could have deteriorated since the onset of the pandemic.Health inequalities may have increased along the axes of social class, gender, age, ethnicity, and territory as a result of COVID-19, and may be even greater in the mid-term compared with the short-term.Chronicity and resulting disability may have increased significantly since the beginning of the pandemic.Since the beginning of the pandemic, the care burden has increased significantly for women in the short-term and this might have had a highly negative impact on their health and well-being.Social and emotional support in the population aged over 55 years might have decreased significantly since the beginning of the pandemic, with the greatest differences perhaps being observed in single-person households in urban areas.

### 1.4. Objectives

General: To determine the magnitude, characteristics, and evolution of the impact of COVID-19 on overall health and its socioeconomic, psychosocial, behavioural, occupational, environmental, and clinical determinants in the general population, and that with greater socioeconomic vulnerability.

Specific:To determine the short- and mid-term impact of the COVID-19 pandemic on the health and emotional well-being of the general population of Andalusia.To analyse the evolution of the socioeconomic, psychosocial, behavioural, clinical, and environmental determinants of health in the context of the COVID-19 pandemic in the population under study.To identify health inequalities along the axes of social class, gender, age, ethnicity, and territory, and their evolution in the context of the COVID-19 pandemic.To evaluate different research sampling techniques to improve the reliability and precision of estimates obtained through surveys using longitudinal designs.

## 2. Materials and Methods

### 2.1. Study Design

This study employs a real-world data design to integrate observational data extracted from multiple sources, including information obtained from different providers based on surveys and clinical, epidemiological, population, and environmental registries.

The surveys have an overlapping panel design [54] to ensure there are both cross-sectional and longitudinal estimates, and to include population-based probability samples. Thus, the ESSOC is made up of a series of measurements broken down into a new sample and a longitudinal sample for each measurement.

### 2.2. Geographical, Population, and Temporal Scopes

The geographical scope is the Region of Andalusia, Spain, and the population scopes are the general population over the age of 16 (ESSOCgeneral), the population residing in disadvantaged areas (ESSOCzones) [55], and the population over the age of 55 (ESSOC + 55). Collective households (i.e., hospitals, nursing homes, barracks, etc.) are not considered in this study. That said, the study sample includes families who, as an independent group, reside in these collective establishments (e.g., a director or janitor of a centre). The temporal scopes of each sample (Figure 1) are:

-ESSOCgeneral: five measurements taken between 2020 and 2022, at baseline (beginning of the Spanish State of Alarm), at one month from the first interview, then at 6 months, 12 months, and 30 months. The first four measurements were carried out from April 2020 to June 2021.

-ESSOCzones: two measurements, taken at baseline (12 months from the beginning of the State of Alarm) and then 12 months from the first interview. These measurements have yet to be performed.

-ESSOC + 55: two measurements, taken at baseline (6 months from the beginning of the State of Alarm), then 24 months from the first interview. The first measurement was carried out in October and November 2020.

The same 30 months in the populations studied (ESSOCgeneral, zones and +55) were skipped to gather information in the mid-term from the State of Alarm in Spain being initiated, as well as to compare their results in the same timepoint.

### 2.3. Sampling Frame

The first measurements are different for each ESSOC survey depending on the availability of the sampling frames. Therefore, ESSOCgeneral was the first survey to be carried out because the sample frame was already available, while the ESSOC + 55 sampling frame had to be developed from scratch and the ESSOCzones sampling frame had to be integrated into the same ESSOC general sampling frame to geographically identify the population located in disadvantaged zones. These areas are defined according to social (unemployment, immigration or social disintegration), urban planning (housing) and educational (illiteracy, absenteeism or school failure) criteria [55].

Thus, the sampling frame used to extract the ESSOCgeneral and ESSOCzones samples has been obtained from the Longitudinal Andalusian Population Database (BDLPA, Base Longitudinal de Datos de Población de Andalucía) [56]. The BDLPA originates from integrating data obtained from the Civil Registries with respect to births, deaths, and marriages (i.e., vital statistics (MNP, Movimiento Natural de la Población)), as well as that reported in the population and housing censuses, give rise to an integrated longitudinal frame for population and territorial statistics in Andalusia [57]. The selected samples are linked to the information obtained from the User Database (BDU, Base de Datos de Usuarios) [58] of the Andalusian Public Health Care System in order to obtain the telephone numbers of the people selected (sample units). The BDU coverage in terms of contact telephone numbers for the selected samples is usually above 96%. On the other hand, the ESSOC + 55 sampling frame corresponds to the user population of the Andalusian Guadalinfo public network aged 55 years and over [59].

### 2.4. Sample Size

The ESSOC is made up of a series of measurements for each population study group (general, zones and +55) with a total of 22,000 effective interviews being carried out over three years. As of June 2021, almost 13,000 effective interviews have been conducted.

During the first measurement (M1), the ESSOCgeneral scope started with a random ‘teorical’ sample comprised of 5000 people. This sample includes respondents and non-respondents for whatever the reason for not responding. Its sample size was calculated under the assumptions of maximum variability in the estimate (p = q = 0.5), a precision of 2.4 percentage points for estimates in Andalusia, a confidence level of 95%, a non-response rate of 40% (whatever the reason) and a design effect of 1.8. The subsequent measurements (M2–M5) are comprised of the longitudinal theorical samples of the previous measurements and, in addition, of a new theorical sample in each measurement. That new theoretical sample is selected according to the design of the first measurement, except for M5 which will incorporate a new stratum of ‘residing or not in disadvantaged areas’ [60]. Finally, due to non-response, from these longitudinal and new theoretical samples an ‘effective sample’ is obtained for each measurement. Thus, those effective samples include just the respondents. With respect to the theoretical sample size for the longitudinal sample of a measurement, this is defined by the effective longitudinal sample and the effective new sample from the previous measurement. To maintain the effective sample size provided by the first measurement of the ESSOCgeneral, the aim for the subsequent ones is to reach an effective sample about 3000 people per measurement, while the effective sample size for ESSOCzones has been established at 2750 people and at least 1350 for ESSOC + 55, assuming theoretical response rates of 60% (considering all types of indidents).

Sample size for the first measurement of the ESSOCgeneral was estimated according to a non-response rate of 40% (Figure 1). This response rate was considered as the percentage of surveys carried out or effective sample with respect to the total of the theoretical sample. Thus, this calculation includes in the denominator all types of incidents, that is, not only those directly related to the refusal of the selected person to be interviewed, but also to the following incidents:from the sampling frame (e.g., incorrect or unreachable telephone number, or person residing in a place other than the one selected);when conducting the interview (e.g., person who cannot be reached, appointment outside the fieldwork time, or insufficient number of attempts to finish the fieldwork, that is, most would be cases that would have answered the interview if it had been extended to the end of the fieldwork);leaving the cohort (e.g., deceased, long-term hospital admissions, change of residence etc.).

The reason for including all types of incidents in the response rate is to obtain a sufficient theoretical sample size to reach the effective sample size of 3000 interviews. This means that, once almost all the fieldwork of the longitudinal sample had been completed, a new sample was added until the total effective size of 3000 interviews was reached. When that sample size was achieved, the field work was completed at each measurement.

In short, if we take into account only the refusals of the selected person, our total response rates are above 70%. Moreover, the field work carried out has introduced elements of improvement that have allowed for an increase in the response rates from one measurement to the next, both for the longitudinal samples and for the new ones that were incorporated in each measurement.

Thus, for measure 1 of the general ESSOC, the response rate was 88.5%. For measure 2, it was for the longitudinal sample 78.3% and for the new one 58.2% (total 73.0%). For measure 3, it was 88.8% and 75.3%, respectively (total 83.1%), and for measurement 4 it was 91% and 78.2% (total 87.8%, Figure 1). 

### 2.5. Sample Selection and Sample Allocation

The probability sampling method used in each measurement to obtain the ESSOC general sample is the stratified random sampling. Twenty-four strata are defined by eight provinces and three degrees of urbanisation (urban, intermediate density, and rural area) [61] (crossed stratification). Allocation of the new samples for each measurement is proportional to the population size of each stratum of province and degree of urbanisation. This implies that within each stratum (province-degree of urbanisation), any person has the same probability of being selected, that is, self-weighted samples are obtained in each stratum.

In addition, for the M5 measurement, the distribution of the new sample will be performed, in the first place, in the two DA strata and, subsequently, as in the case of the previous measurements.

The theoretical longitudinal sample of a measurement is composed of the effective sample of the previous measurement, except for measurement M1 which, being the first one, does not have a longitudinal sample.

In the ESSOC + 55 sample, the first measurement was stratified by clusters (Guadalinfo Centers, N = 651), with sub-sampling to 1200 users. These centres are stratified per Andalusian province and inhabitation level (<10,000 inhabitants, 10,000–19,999, ≥20,000), as well as sex and age quotas (55–64 years, 65–74 years, and >75 years). As in the case of the ESSOCgeneral sample, the ESSOC+55 second measurement will be made up of a longitudinal sample (that of the first measurement) and another new sample, until a total of 1200 interviews are completed.

### 2.6. Fieldwork

The survey information is collected through a computer-assisted telephone interview (CATI). The management, control, and monitoring of the collection of information is carried out through Pl@teA and MobiNet Gandia Integra software. The data collection is carried out by a team of between 8 and 12 interviewers assigned solely to this study. This ensures working team stability, which is of fundamental importance in regular, longitudinal surveys. Before starting the study, the interviewers will receive the necessary training regarding the content of the survey. To this end, in addition to virtual meetings held before starting the fieldwork, they are provided with interviewer’s and questionnaire manuals which, besides explaining the questionnaire and study’s content, describe the survey platform, possible incidences, and the protocols to be followed in each case to guarantee the maximum quality of the samples and the information collected. Prior to engaging in the fieldwork, each interviewer performs several pilot tests to measure times and determine the complicated points of the questionnaire.

The schedule set to conduct the interviews is from Monday to Friday from 10:00 to 21:00, and on Saturdays from 10:00 to 15:00 for the first measurement, and for the rest of the measurements from Monday to Saturday from 14:00 to 21:00, although deferred appointments can be scheduled without time limits. Furthermore, a telephone line with a 900 prefix and staffed by telephone agents is made available to the public. This number is provided to the survey holders via text message or through the CSyF website, where the characteristics of this study are also published. The call centre also receives calls from people who, after having been contacted by CATI agents, need to confirm the official nature of the survey. In fact, many of these calls become completed interviews.

### 2.7. Quality Control

For the ESSOCgeneral, quality control measures mean the interviews are cross-checked both internally in the call centre itself and in the IECA and EASP. Each interviewer is monitored to ensure that they follow the established protocols and that they use each type of incidence correctly. The intervals elapsed between each call and their duration are also monitored.

In addition to recording the calls made by each interviewer, a ‘listen-in’ check is also performed to review both the positive aspects and those to be reinforced in the supervised interviews (i.e., for 10% of the calls performed for the first measurement and 25% of the calls performed for the rest of the measurements). During these checks, aspects, such as the interviewer’s self-presentation, their presentation of the study, providing the 900-prefix telephone number, confirming the place of residence, the correct delivery of the questionnaire’s questions, and all response options, are assessed.

Quality control, data cleansing, and data coding are carried out simultaneously with the fieldwork with the aid of the software to be used in the study. Each interviewer is provided with a space on the platform to record observations during the survey being conducted. This then allows the supervision team to cleanse those interviews in which the interviewer detected an inconsistency in the respondent’s answers, or those in which the interviewer made a mistake when completing the questionnaire. Likewise, the values of the variables are revised, and invalid ones cleansed. Moreover, the coding of the variables corresponding to open-ended questions, such as the respondent’s occupation and educational level, is carried out in tandem with the fieldwork. In the rest of the open-ended questions, prior to their coding, their possible answers are cleansed, and the categories deemed to correspond to the majority subsequently coded.

During the telephone interviews, different situations may arise that could result in the inability to complete the survey. These are known as field incidences (Table 2), with the most important types being final incidences, i.e., those that, after several attempts, finally result in the inability to complete the survey.

As this is a longitudinal study, one of the most significant reasons for a lack of response is if the potential interviewee identifies the incoming call number and does not answer the phone. To solve this, as much as possible, the telephone number from which each call is made is changed periodically, so that even if a number were to be identified and blocked, we could continue to attempt to contact that person by employing a new telephone number.

In addition to the quality of the sample, there are other factors of interest in assessing how fieldwork has developed during a surveying operation. One such factor is to determine how a survey has been carried out in terms of effectiveness and efficiency. The most direct way to measure this is to calculate the number of attempts or calls that had to be made to complete each survey. This type of information is also very useful to be able to design strategies aimed at optimizing attempts and, therefore, increasing sample levels in future operations.

### 2.8. Sampling Weights

The original sampling weight for the new samples is obtained from the inverse values of the effective sampling fractions in each stratum and used to calculate the Hajek estimator [62]. This is subsequently calibrated to obtain more reliable estimates based on the demographic characteristics of Andalusia. To this end, we use a truncated linear calibration method [63] and, as auxiliary information, the marginals of the Andalusian population per sex and age (16–19, 20–24, 25–29,..., 75–79, and ≥80 years old), sex and province, sex and nationality (Spanish or dual nationality and foreign), and sex and degree of urbanisation. These data are obtained from the Continuous Municipal Register (Padrón Continuo de Municipios) [64]. Regarding the non-response bias in longitudinal samples, we can predict non-contact and non-cooperation based on auxiliary information and information already known about the sample subjects. Thus, the original weights used in the estimates of longitudinal sample M_t are corrected during a first phase by modelling the non-response with respect to the longitudinal effective sample obtained in M_t-1 using machine learning techniques [65]. Said non-response is estimated using a XGBoost model [66], which represents the state-of-the-art in machine learning. Every piece of data and variable from the M_t-1 sample is used for training; thus, the algorithm has all the information available to learn. Likewise, the hyperparameters of the model are optimized using cross-validation to ensure reliable estimations. Then, during a second phase, they are calibrated following the method described for the new samples. As auxiliary variables, we use those extracted from the Continuous Municipal Register (e.g., nationality, sex, age, province, degree of urbanisation, etc.) and the registers from the ESSOC itself in M_t-1 (Table 3).

In addition to these adjustments, other methodological alternatives, not yet explored in this type of sample design, for instance, double calibration, will also be investigated by considering different variables in order to model non-responses and, on the other hand, correction of the representativeness bias [67,68] and machine learning techniques, and adjusting non-responses with the aid of the Propensity Score Adjustment (PSA) [69,70].

### 2.9. Variables

The study variables will mainly be extracted from the following sources: BDLPA; the Andalusian Population Health Database (BPS, Base Poblacional de Salud) [71]; the Andalusian Environmental Information Network (REDIAM) [72]; the Andalusian Epidemiological Surveillance System (SVEA) [73] and the ESSOCgeneral, ESSOCzones, and ESSOC + 55 surveys.

The personal data of the participants selected for the interview (name, surname, and telephone number) are extracted from the BDLPA. In addition, the BDLPA is linked annually with a repository of georeferenced buildings so that the postal address and coordinates (250 m × 250 m grid) in the territory can be extracted. This will allow us to extract geographical factors (urbanisation degree and population density, among others) via other IECA registries, and environmental factors (pollution and temperature, among others) via the REDIAM registry from the Andalusian Regional Government’s Department of Agriculture, Livestock, Fisheries, and Sustainable Development (Consejería de Agricultura, Ganadería, Pesca y Desarrollo Sostenible de la Junta de Andalucía).

From the SVEA registry, epidemiological information related to COVID-19, such as the date and result of the diagnostic test for active infection (PDIA), will be extracted to detect the presence of an active SARSCoV-2 infection, which includes both reverse transcription–polymerase chain reaction (real time RT–PCR) as the antigen (Ag) rapid test; date of the onset of symptoms; close contact of confirmed case with PDIA; local or imported case; occupation as a health or social health professional; need for hospitalisation or admission to an intensive care unit; date of admission, and discharge.

In addition, the clinical information related to chronic diseases [74], functional and cognitive assessments, health resources (volume and cost), population stratification, and drugs consumed, which is obtained from the BPS, will also be added to the valid samples (Table 4). Further information about the variables and the main features of the abovementioned registers can be found in Appendix A.

With regard to the surveys, each measurement is associated with a questionnaire that coincides, to a significant extent, with previous measurements to enable an analysis of its evolution, and to incorporate new information to analyse specific characteristics present at each moment of the pandemic. Repeating unchanging information in subsequent measurements is avoided in the case of longitudinal samples. The questionnaire used for each measurement is organised into blocks of information, as shown in Table 5.

### 2.10. Data Analysis

The analyses take advantage of all the information available from the measurements and the auxiliary information sources and will be carried out with the free software environment R [81], considering the sample design, as well as the calibration and inference methods described in the previous sections. The use of free software will guarantee transparency and facilitate the replicability of the study.

A table will be prepared for each variable of each measurement, together with the variable’s original response categories, including the valid sample size (*n*), the percentage of lost samples, the population size estimate (N), the relevant statistic (mean or percentage), the 95% CI, and the coefficient of variation (CV), for both the total and the cross-disaggregation per sex and age (16–29/30–44/45–64/65+), as well as per province and urbanisation degree. The sample size is recorded for the total and the categories of the segmentation variable. In the case of cells with CV estimates >20%, the CV will be indicated in a footnote to the table.

The variables shared by all measurements are dichotomized based on the results reported in the previous tables identifying, in each case, the most convenient category to be highlighted based on the previous tables. In addition, a table describing the specific estimates with their corresponding CV will also be created.

Alternatively, to evaluate the changes in each measurement with respect to the first one, both the population affected by the change and the percentage segmented per demographic and territorial variables will be estimated. The *p* value will be calculated to evaluate the effect of such change and will be indicated in a footnote to the table using three categories: *p* < 0.001, *p* < 0.05, and *p* < 0.1.

In the case of variables that coincide in each pair of consecutive measurements (M2–M1, M3–M2, and M4–M3), the estimated percentage of the difference between one measurement and the previous one will be calculated as well as the estimate of the population size and the signalling when the CV is greater than 20% and segmented by the demographic and territorial variables.

To analyse factors associated with variables of a given measurement or variables measuring the change between one measurement and another, we will use multivariate explanatory models adapted to the characteristics and nature of the variables and specified as generalized linear mixed models (GLMM) with a family dependent on the type of dependent variable: Gaussian, when the variable is continuous (equivalent to a linear regression); binomial, when the variable is dichotomous (equivalent to a logistic regression); or Poisson, when the variable is discrete (equivalent to a Poisson regression). Random effects will be included in these models with two goals: first, capture the effects of unobserved confounders, and second, capture the time dependency (as you will get repeated measures of the subjects), and the spatial dependency (i.e., spatial clustering), if applicable. Inferences will be made following a Bayesian perspective and using the integrated nested Laplace approximation (INLA) [82,83]. We will use penalised complexity priors known as PC priors. These priors are robust in the sense that they do not impact the results and, in addition, they allow for an epidemiological interpretation [84]. The analyses will be carried out using free software R (version 4.0.4 or greater) [81] through the INLA package [82,83,85].

Finally, advanced data visualisations will be used to allow an in-depth exploratory analysis of the evolution of the study variables and a representation of the main results of the produced models. These visualisations will be developed using Python [86] programming language and integrated into software and web solutions that allow for interaction and dissemination.

### 2.11. Data Management Plan

The data management plan is provided in Appendix A. The type and format of data that will be collected and generated within the scope of this project is described in this plan, together with the procedure provided to access data (by whom, how, and when it can be accessed), data ownership, repository to deposit data, and procedure planned to guarantee the specific ethical and legal requirements.

Details of the Data Protection Impact Assessment (DPIA) will also be presented here in accordance with the specific adaptation of this methodology to research projects in the health care sector [87,88,89]. Thus, the need for a DPIA was confirmed from the outset (Appendix A). Subsequently, the data lifecycle was defined (Appendix A), and the need and proportionality of the processing were analysed (Appendix A) and, finally, a risk assessment and action plan developed (Appendix A).

## 3. Discussion

Given the characteristics, current and future repercussions of the current pandemic, developing this research project will make it possible to periodically obtain relevant information for decision-making processes in social and health matters and, therefore, promote a more efficient, reliable, and responsible science for social change (such as the one that we are currently experiencing as a result of the pandemic). In addition, it will encourage:institutional alliances of great social value between the public administration, health care services, and the scientific and academic community;powerful and novel methodologies in the fields of public health, epidemiology, and sampling to reduce potential biases caused by a lack of coverage, response, or non-randomised selection [1,50];large, integrated, quality, and open databases containing information extracted from clinical and non-clinical population registries; data concerning social, economic, and environmental contexts and the perception of the population, along with foreseeing the future incorporation of genomic information [90];the systematic review throughout the entire project of the scientific evidence obtained through this type of study;training with a view to transmitting the available knowledge and increasing capacities and skills in designs, sources, and methodologies;measuring the short- and mid-term impact of COVID-19 at different times and on different populations since the beginning of the official State of Alarm.

The limitations of the study include those derived from the coverage and quality of the sampling frames, which may cause selection biases. In the case of the BDLPA, its telephone coverage is over 90% and it tends to have low percentages of non-existent telephone numbers (7–8%) or non-contactable telephone numbers due to the frame’s outdatedness (9–10%). That said, any potential biases due to such defects will be corrected through the estimator calibration techniques described above. Another limitation is that caused by the longitudinal design of the ESSOC surveys in terms of panel attrition, which could lead to potential biases due to an absence of response. We estimate the response rate for the full longitudinal sample to be 25–35% (the effective sample in M5 of the effective sample in M1 is divided by the effective sample in M1). To reduce the effect of potential non-response biases, we decided, on the one hand, to use an overlapping panel design as this type of design allows for completing each measurement with new cross-sectional samples that become longitudinal in subsequent measurements and, on the other hand, to make adjustments in each measurement according to the weights of the longitudinal samples.

This study was initially designed to also include population from nursing homes, but ultimately this was not possible. This is an unfortunate limitation of the study considering the enormous impact that COVID-19 has had on this population. In addition, this project may have information biases intrinsic to sampling research. Consequently, we have chosen to employ scales and variables widely used in population-based health surveys and that also allow comparisons to be made for most of the key indicators. Finally, as in most of these types of studies, its feasibility and limitations will, to a large extent, depend on the quality and availability of the data from different data holders. Data availability for this project would appear not to be a limitation, given that the request for the processing of these data is in accordance with the policies on data disclosure for processing data in research projects in the public sector. Additionally, the EIPD performed resulted in an acceptable residual risk level (see Appendix A). This evaluation highlights, among other elements, that the data provided is protected by statistical confidentiality, that it is not misused, that its treatment is anonymous and global at all times, and that indirect identification is impossible. Furthermore, the project’s results will be beneficial to the general population in a holistic way, thanks to its socioeconomic and environmental context, and its evolution over several years from the onset of the pandemic. Therefore, the risk to the privacy of the study population is minimal compared to the potential benefits the results will bring. The transfer of data will be carried out between organisations within the Junta de Andalucía, as this will be within the context of a research project exclusively in the public sphere and under the legitimate use of records as research infrastructures in accordance with the Spanish health legislation (see Appendix A). With respect to data quality, the reproducibility of this research will be conditioned by the way in which the data have been obtained, a phenomenon common to all RWD studies. Even when these are the best possible, the fact that the data are extracted from official sources does not guarantee their quality. Due to this, the owners of the information will be asked to describe the detailed procedure used to extract the data, as well as any previous processing it may have undergone. It should be noted that individuals collaborating on this project also belong to the work teams of the main sources of information to be used which, therefore, should guarantee success in the interpretation of the original data and the optimisation of the extraction strategies.

## 4. Conclusions

The ESSOC will enable precise and valid analysis of the short- and mid-term impact of the policies applied, and interventions made to, not only of the health of the general population, but also the most vulnerable population, during the pandemic. The study will also determine the evolution of the socioeconomic, psychosocial, behavioural, occupational, environmental, and clinical determinants of health, and identify the inequalities in health in all its axes (i.e., social class, gender, age, ethnicity, and territory).

The conceptual approach of this study will encompass all aspects affecting health and so will contribute to an extraordinary increase in the current knowledge concerning the impact the COVID-19 pandemic is having. This knowledge will, in turn, be crucial for health systems to be able to design quick and effective interventions aimed at improving the health care, health, and quality of life of the populations most affected by the COVID-19 pandemic. Moreover, the project management model based on collaborative, multidisciplinary, and open research will allow the critical mass needed to be generated to thus achieve the objectives that have been set (i.e., at the populational level, as well as at an individual and disease level, in our case, COVID-19).

## Figures and Tables

**Figure 1 ijerph-18-08120-f001:**
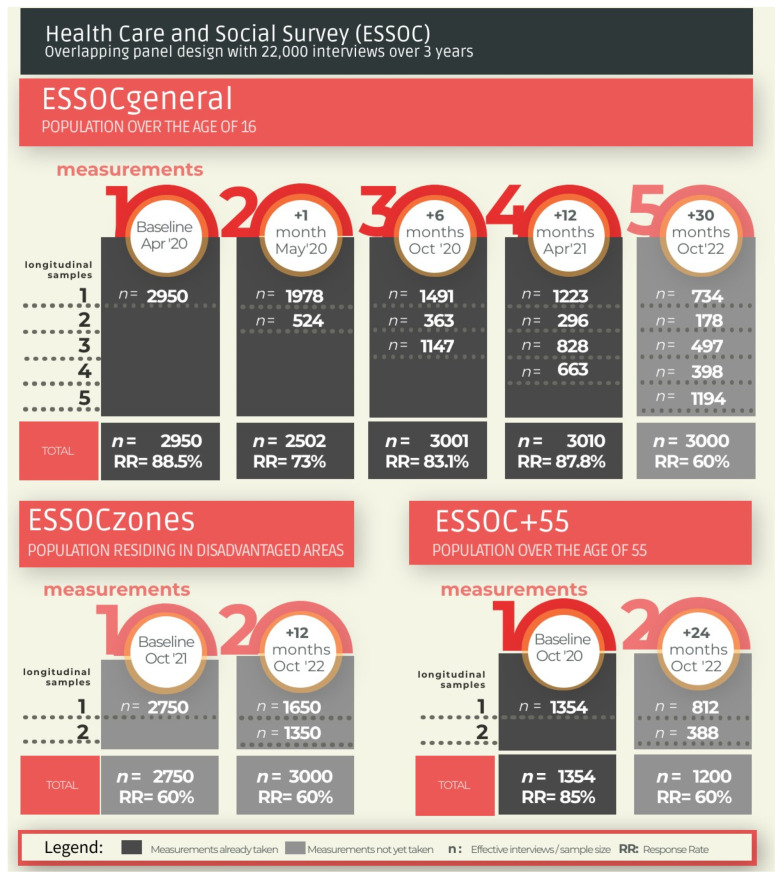
Health Care and Social Survey (ESSOC): Population and Temporal Scope, and Samples based on an overlapping panel design (Response Rates of the measurements already taken are calculated only with the negatives, the rest of them are calculated with all kinds of fieldwork incidences).

**Table 1 ijerph-18-08120-t001:** Health Care and Social Survey (ESSOC): Study Framework.

Focus	Design	Dissemination	Management
Public Health	Real-World Data	Open Science	Collaborative and Multidisciplinary Research
Community participation to obtain information on people’s health and quality of life to be able to intervene both individually and collectively in the face of the pandemic	Integration of observational data extracted from multiple sources from different perspectives based on probabilistic samples and administrative registries	Results, methodologies, processes, and collected data distributed, reused, and freely and openly accessible	Andalusian School of Public Health, Andalusian Health Service, Department of Health and Families (Andalusian Regional Government), Universities of Granada and Girona, Andalusian Institute of Statistics and Cartography and Guadalinfo Network

**Table 2 ijerph-18-08120-t002:** Health Care and Social Survey (ESSOC): Interview incidences and protocol to be followed:

Incidence	Incidence	Description	Protocol
Frame incidence (reasons that make it impossible to complete the survey due to problems related to the sampling frame; for example, a telephone number with which to contact the sample person could not be obtained or the housing frame was not sufficiently up to date).	The telephone number does not exist.	Wrong number: the telephone number dialled does not exist, corresponds to a fax, or has restricted calls.	Direct removal.
	Not contactable.	Out-of-date frame: the selected person is living in a different municipality, a telephone frame without a telephone number, a person unreachable through the telephone number/home address provided due to circumstances such as death, divorce/separation, etc.	Direct removal.
Relationship-situation incidences (reasons that make it impossible to complete the survey due to several types of situations affecting the surveyed people, for instance, they cannot be located, they refuse to participate in the survey, or any other aspect that prevents the survey being conducted).	No contact.	The household cannot be contacted (e.g., nobody answers the telephone, or the answering machine goes off).	Removal after four attempts performed on two different days, at two different times.
	Absent.	The selected person cannot be contacted.	Removal after four attempts performed on two different days and at two different times.
	Inability to answer.	The selected person cannot complete the survey due to an inability to respond to it because of disability, age, illness, lack of knowledge of the language, or any other circumstance. If possible, the survey should be completed by a close relative.	Direct removal.
	Refusal.	The selected person refuses to complete the survey or refuses to continue it after it has begun.	Direct removal.

**Table 3 ijerph-18-08120-t003:** Health Care and Social Survey (ESSOC): Adjustment of the design sampling weight in each measurement.

	Type of Adjustment	
Sample Type (Effective)	1st Phase	2nd Phase
New	Non-response adjusted by proxy based on the effective sample size in each stratum.	Representativeness by truncated linear calibration with 0.1 and 10 limits based on the auxiliary variables
Longitudinal	Non-response adjusted using an XGBoost model based on variables from the previous measurement.	

**Table 4 ijerph-18-08120-t004:** Health Care and Social Survey (ESSOC): Auxiliary sources and variables.

Registry	Description	Information	Variables Extracted
BDLPA–Longitudinal Andalusian Population Database [71]	Information from the census coordination system and civil registries that give rise to a consolidated framework of the Andalusian population	Personal data	Name, surname, identification health number (NUHSA), geographical coordinates
BDU–User Database of the Andalusian Public Health Care System [58]	Contact Information of the Andalusian Public Health Care System	Personal contact information	Telephone numbers
BPS–Andalusian Population Health Database [71]	Personal health information from the Andalusian Population Health Database and Health care information	Health and health care information	Chronic diseases, functional and cognitive assessments, health resources (volume and cost), population stratification, drugs consumed
REDIAM–Andalusian Environmental Information Network [72]	Daily averages by collecting/meteorological station and at the census section level	Pollution, temperature	Mean daily values from pollution, air quality and temperature
SVEA–Andalusian Epidemiological Surveillance System [73]	Functional organisation for health surveillance that collects, among other things, epidemiological information related to SARS-COV-2 infection	Epidemiological information of COVID-19	PCR results, symptoms’ date, close contact, health care professional, hospitalisation unit (specifying ICU), date of admission and discharge, need of mechanical ventilation and clinical data

For further details, see Appendix A.

**Table 5 ijerph-18-08120-t005:** Health Care and Social Survey (ESSOC): Information blocks and variables entered in each measurement.

Subject Area	1st Measurement (M1)	2nd Measurement (M2)	3rd Measurement (M3)	4th and 5th Measurements (M4 and M5)
Household and housing characteristics	Municipality, usual household, type of household, surface area, facilities, household changes, number of cohabitants (<6/<16/>60), type of household, and equipment.	Municipality, usual household, type of household, surface area, facilities, household changes ^b^, number of cohabitants (<6/<16/>60), equipment, number of rooms, and number of inhabitants with disabilities or requiring care.	Municipality, usual household, type of household, surface area, household changes, number of cohabitants (<6/<16/>60), number of rooms, and number of inhabitants with disabilities or requiring care.	Municipality, usual household, type of household, surface area, household changes, number of cohabitants (<6/<16/>60), number of rooms, and number of inhabitants with disabilities or requiring care.
Time use and cohabitation	Household chores, care tasks, daily activities during the confinement period (at home and outside), cohabitation and relationships, and causes for optimism.			Household chores, care tasks, daily activities ^b^ during the confinement period (at home and outside), cohabitation and relationships, and causes for optimism.
Health and emotional well-being	COVID-19 diagnosis, severity, diagnosis within the person’s settings, self-perception of general and mental health (current and last year), emotional well-being ^c^, difficulty to withstand the confinement, malaise, chronic illness, and change of medication.	COVID-19 diagnosis, severity, diagnostic tests, diagnosis within the person’s settings, self-perception of general and mental health, emotional well-being ^b^, cohabitation, difficulty to withstand the confinement, happiness, social and emotional support ^c^, malaise ^b^, chronic diseases (suffering and limitations), and medication (use and change of use ^b^).	COVID-19 diagnosis ^b^, severity, diagnostic tests, diagnosis within the person’s settings, self-perception of general and mental health, emotional well-being ^b,c^, happiness, social and emotional support ^b,c^, malaise ^b^, chronic diseases (suffering and limitations), and medication (use and change of use ^b^).	COVID-19 diagnosis ^b^, severity, diagnostic tests, diagnosis within the person’s settings, self-perception of general and mental health, emotional well-being ^b,c^, happiness, social and emotional support ^b,c^, malaise ^b^, chronic diseases (suffering and limitations), and medication (use and change of use ^b^).
Habits and lifestyle	Habit modification (exercising, smoking, alcohol consumption, sleep, and diet).	Habit modification ^b^: exercising, drinking, smoking, sleep, food, daily intake of vegetables and fruit, exercising, weight and height ^c^, smoking, alcohol consumption, sleep, and flu vaccination.	Habit modification ^b^: exercising, drinking, smoking, sleep, food, daily intake of vegetables and fruit, exercising, weight and height ^c^, smoking, alcohol consumption, sleep, and flu vaccination ^b^.	Habit modification ^b^: exercising, drinking, smoking, sleep, food, daily intake of vegetables and fruit, exercising, weight and height ^c^, smoking, alcohol consumption, sleep, and flu and COVID-19 vaccination ^b^.
Economic situation and socio-demographic characteristics	Educational level, employment situation, working from home, type of contract, occupation ^c^, cohabitation with a partner, identification of the cohabitant with the greater income (educational level, employment situation, type of contract, occupation), difficulty in making ends meet, late payments, income, future worries, and degree of confidence in public institutions.	Employment situation, educational level, occupation ^c^, development ^b^, ability to work, identification of the cohabitant with the greater income (educational level, employment situation, occupation), difficulty in making ends meet, late payments, change in economic situation, parents’ educational level, and future worries.	Employment situation, educational level, occupation ^c^, development, ability to work, identification of the cohabitant with the greater income (educational level, employment situation, occupation), difficulty in making ends meet, late payments ^b^, change in economic situation, parents’ educational level, and future worries.	Employment situation, educational level, occupation ^c^, development, ability to work, identification of the cohabitant with the greater income (educational level, employment situation, occupation), difficulty in making ends meet, late payments ^b^, change in economic situation, parents’ educational level, and future worries.

^a^ Questionnaires are provided in Appendix A. ^b^ Variables that present modifications in their temporal scope in relation to the previous measurement. ^c^ Composite variables: emotional well-being [75], social and emotional support [76,77], body mass index [78], social class [79,80].

## Data Availability

The data, source codes, and other documentation developed in the context of this study will be available through the web platform http://researchprojects.es/ (accessed on 28 July 2021), from the open source project management tool https://osf.io/ (accessed on 28 July 2021) and from the Oxford Supertracker global directory of policy trackers and surveys related to COVID-19 https://supertracker.spi.ox.ac.uk/surveys/ (accessed on 28 July 2021).

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
