# Peer review of "Impact of COVID-19 on the Health of the General and More Vulnerable Population and Its Determinants: Health Care and Social Survey–ESSOC, Study Protocol"

_ijerph, 2021, doi:10.3390/ijerph18158120_

Round 1
Reviewer 1 Report
Sanchez-Cantalejo et al present a study protocol for a health care and social survey. The study uses a probabilistic, overlapping panel design to explore the impact of COVID-19 on the health and its determinants in both the general and vulnerable populations of Andalusia.
This is an original and ambitious study on a topic of very high importance.
Minor comments:
# Introduction
1) The background section would perhaps benefit from some restructuring. Could the authors group more clearly the evidence of the impact of the epidemic on the elderly, those with chronic diseases, ethnic minorities and migrant population? Sub-headings for each group may help. can the authors shorten this Background section a little? For example, perhaps it is not necessary to give as much information as is provided in the manuscript for the National Seroepidemiology study.
2) In the Objectives section, it is not clear if there are 4 or 5 objectives. Please clarify
#Materials and methods
1) The authors should describe the methods more clearly. Some pointers:
-Please be more explicit about the date when the study was started and which measurements have already been done for the survey. How many interviews are there so far?
-I think Figure 1 needs a little more explanation. For example, stating somewhere in the figure that the numbers in the columns are the number of telephone interviews in each sample, or explaining why some of the columns are in black colour and some in grey colour (is it because for the columns in black the telephone interviews have already happened and the relevant data has already been collected?).
-Can the authors briefly explain early in the manuscript the rationale for having a different number of measurements for the different components of the surveys (ESSOCgeneral, ESSOCzones, ESSOC+55)?
-In the sample size section, it would be helpful to define theoretical sample and effective sample at the beginning and clearly state what are the differences between the two. Also it would be helpful if the authors could discuss a little about the relevance of the sample size. Also it would be helpful if the authors make more explicit '(perhaps earlier in the manuscript) the rationale for generating new samples at each measurement of the survey. Finally, it seems to me like the last sentence in this section could go at the beginning of the section
-In the sample allocation section, the sample units are not clear. What is the sample unit? Municipality? Other? Also It is not entirely clear how the sample allocation is done. What do the authors mean with uniform allocation by province? How many sample units are there per province? The use of the term "new sample" at the beginning of the section is somewhat confusing. I presume that "new sample" refers to all samples from all measurements. Can the authors clarify?
-In the sample selection section, there is no mention of the sample units. It would be helpful to refer to these when describing the sample selection
-In general, can the authors describe sections 2.2-2.6 of the Materials and Methods using a flow diagram? I think this would add clarity to the description of the sampling
-In the scoping review section, it is not clear what is the "final objective" that the authors refer to. Please clarify this
I would strongly suggest that the authors perform an interim analysis of the data to identify potential problems and solutions. This study could be published.
Finally, kindly check the English language and style. For example, in the first sentence of the last paragraph of section 2.7, "the schedule set to conduct the surveys" might be better expressed as "the schedule set to conduct the interviews".
Author Response
Dear Reviewer,
We would like to thank you for your interest in our article and for the time that you have devoted to reviewing it.
We are pleased to submit a revised version of our manuscript. We have highlighted the changes by using the track changes mode in MS Word. As detailed in our responses, we have made changes in the manuscript to address the shortcomings identified by you.
Below we respond to your comments:
# Introduction
COMMENT: 1) The background section would perhaps benefit from some restructuring. Could the authors group more clearly the evidence of the impact of the epidemic on the elderly, those with chronic diseases, ethnic minorities and migrant population? Sub-headings for each group may help. can the authors shorten this Background section a little? For example, perhaps it is not necessary to give as much information as is provided in the manuscript for the National Seroepidemiology study.
RESPONSE: Thank you for your suggestions. As suggested, three subsections - one for each group of vulnerable population studied- have now been included and the length of the Background section has been shortened by reducing the amount of information from the National Seroepidemiological study.
COMMENT: 2) In the Objectives section, it is not clear if there are 4 or 5 objectives. Please clarify
RESPONSE: There were five objectives but, at the request of another reviewer, we decided not to include the last one related to the scoping review. Thus, we have also eliminated the 2.12 Scoping Review section. The main reason for omitting it from the manuscript is that we believe it is separate from the rest of the paper. Thus, there are four objectives. We hope you agree with this decision as well.
#Materials and methods
1) The authors should describe the methods more clearly. Some pointers:
COMMENT: -Please be more explicit about the date when the study was started and which measurements have already been done for the survey. How many interviews are there so far? I think Figure 1 needs a little more explanation. For example, stating somewhere in the figure that the numbers in the columns are the number of telephone interviews in each sample, or explaining why some of the columns are in black colour and some in grey colour (is it because for the columns in black the telephone interviews have already happened and the relevant data has already been collected?).
RESPONSE: We have added your examples in Figure 1 and it is clearer now. Thank you for your suggestions. In addition, we have added lines to better identify the longitudinal samples and have also updated it with the fourth ESSOCgeneral measurement which was recently performed. We also included in the text which measurements have been already carried out (lines 218-225) and how many interviews there have been so far (line 266).
COMMENT: Can the authors briefly explain early in the manuscript the rationale for having a different number of measurements for the different components of the surveys (ESSOCgeneral, ESSOCzones, ESSOC+55)?
RESPONSE: We have now included an explanation at the beginning of the Sampling frame section (lines 238-243).
COMMENT: In the sample size section, it would be helpful to define theoretical sample and effective sample at the beginning and clearly state what are the differences between the two. Also it would be helpful if the authors could discuss a little about the relevance of the sample size. Also it would be helpful if the authors make more explicit '(perhaps earlier in the manuscript) the rationale for generating new samples at each measurement of the survey. Finally, it seems to me like the last sentence in this section could go at the beginning of the section
RESPONSE: Thank you for your comments. We have introduced changes in the text to clearly state the differences between the theorical and effective samples and to briefly discuss the relevance of the sample size (lines 267-290). We have also implemented some of those changes in Figure 1 as detailed in the previous comment. With regards to the rationale for generating new samples at each measurement of the survey, this has been explained earlier in the 2.1 Study Design section, and we moved part of the last sentence of the Sample Size section to the beginning of this section (lines 264-266) and the other part to Section 2.1 (lines 205-206).
COMMENT: -In the sample allocation section, the sample units are not clear. What is the sample unit? Municipality? Other? Also It is not entirely clear how the sample allocation is done. What do the authors mean with uniform allocation by province? How many sample units are there per province? The use of the term "new sample" at the beginning of the section is somewhat confusing. I presume that "new sample" refers to all samples from all measurements. Can the authors clarify?
RESPONSE: The sample units are the study population, i.e., people to be interviewed. Allocation of the new samples for each measurement is mixed. Firstly, they are uniform by province (150 people or sample units per province) and, secondly, proportional to the population size of each stratum of province and degree of urbanization. This implies that within each stratum (province-degree of urbanization) any person has the same probability of being selected, that is, self-weighted samples are obtained in each stratum.
Sections 2.5 and 2.6 have been rewritten, and now they include more information about the probability sampling design. In addition, both sections (Sample Allocation and Sample Selection) have been merged to make them clearer (lines 321-347). We have also used the term ‘people’ or ‘interviews’ instead of ‘units’ throughout the manuscript.
COMMENT: -In the sample selection section, there is no mention of the sample units. It would be helpful to refer to these when describing the sample selection
RESPONSE: We have introduced changes in accordance with your comment (lines 325 and 258-259). Thank you.
COMMENT: -In general, can the authors describe sections 2.2-2.6 of the Materials and Methods using a flow diagram? I think this would add clarity to the description of the sampling
RESPONSE: We think the new Figure 1 gathers the key points from those sections. It has been greatly improved thanks to you and the other reviewers input. For example, the reviewed version states the numbers in the columns as the number of telephone interviews in each sample, explains the meaning of the colors of the columns, adds lines to better identify the longitudinal samples and has been updated with the fourth ESSOCgeneral measurement which was recently performed. We hope this addresses your comment.
COMMENT: -In the scoping review section, it is not clear what is the "final objective" that the authors refer to. Please clarify this
RESPONSE: We referred to the fifth objective, “To compile and systematize the existing evidence regarding the design, sources, methodologies, and topics related to measuring the impact of COVID-19 on health and its determinants through surveys”. Nevertheless, as we mentioned earlier, we have now omitted that objective and its related section, 2.12, from the manuscript.
COMMENT: I would strongly suggest that the authors perform an interim analysis of the data to identify potential problems and solutions. This study could be published.
RESPONSE: We agree with you. In fact, as we said earlier, we have just finished the fourth ESSOCgeneral measurement so we are performing that interim analysis which will also help us to know more about what has happened over the year that has elapsed the state of alarm in Spain.
COMMENT: Finally, kindly check the English language and style. For example, in the first sentence of the last paragraph of section 2.7, "the schedule set to conduct the surveys" might be better expressed as "the schedule set to conduct the interviews".
RESPONSE: We have had the English language and style checked. The changes are highlighted using the track changes mode in MS Word. We have also expressed ‘surveys’ as ‘interviews’ when needed, and other terms such as ‘transversal design’ as ‘cross-sectional design’…
Once again, thank you for reviewing our article and for your comments, which have undoubtedly improved the quality of the manuscript. We hope we have addressed the shortcomings that you identified.

Reviewer 2 Report
As a global comment, the authors should be mindful of runoff sentences. Many of the sentences in the manuscript were too long, with multiple ideas communicated, making it difficult to maintain focus. Please see below for specific comments.
INTRODUCTION
- What are the lifestyle habits that contributed to further stigmatization of the Roma population? (line 69)
- What is the vulnerable population that is referenced in the last sentence of the second paragraph (lines 71-72)?
- Are the populations highlighted in the second paragraph (Roma, infants and adolescents, other vulnerable) all in Spain or is there a different/broader geographic distribution? Why are they authors highlighting these specific vulnerable populations. What about these populations make them vulnerable (in general and in the context of COVID)?
- Authors should provide more detail (statistics) for how COVID and the lockdowns are already known to have impacted people with chronic conditions.
- Considering the limitations of non-probabilistic samples and web surveys described by the authors, what are the substantive scientific benefits to using such modes of data collection (beside the speed at which data can be collected and reported)?
- Can the authors please explain what they mean by “transversal designs”? (line 96)
- What exactly is meant by “continuous information on the evolution of the epidemic” (line 110-111)?
- In line 119, authors should qualify what they mean by “greatest impact”. In previous paragraphs, authors gave examples for how COVID impacted other populations (in different ways). With the over 70 years of age group, are they referring to mortality?
- What is the reason for the sex disparity in COVID-19 mortality (lines122-123)?
- What is the “specific, reliable and timely” data that is being provided by the Health Care and Social Survey? (lines 140-141)
- What do the authors mean when they say, “…it is populated as other European countries such as Austria or Switzerland.”? (line172-173)
- Section 1.3 reads as statements of fact. Authors should re-write this sentence to clarify that these are all testable hypotheses that the data set will answer. Also, these do not seem like hypotheses because some of this was discussed earlier in the introduction as findings from recent research.
- (line 202) What is the timeframe for short and mid-term impact?
- What do the authors mean by psychosocial and behavioral determinants of health (as these can also be health outcomes)? (line 204)
METHODS
- Can the authors provide more background for what is considered a disadvantaged area (line 225)?
- The omission of nursing homes will be a significant limitation to this data, considering the impact of COVID on the elderly (line 227). Do the authors plan to just accept this limitation or is there a way for them to augment the data set with this information to understand the impacts of COVID?
- Why did the authors skip 24 months as a timepoint for data collection? (line 233)
- Why do the residents of the ESSOCzones only have two points of data collection compared to ESSOCgeneral? Considering the presumed vulnerability of the residents in the ESSOCzones, it would be important to collect just as much information on these people as for the ESSOCgeneral. Same comment applies to the ESSOC+55 group.
- Why do the authors expect such a high non-response rate (40%)? (lines 261-262). If the authors are anticipating so much non-response and loss to follow-up, can they modify the study design to improve participation?
- How do the authors plan to account for the mixed samples at each sampling event (one group is maintained from the prior sampling, the other group is newly sampled)?
- “2.5 Sample Allocation” more details need to be provided regarding the sampling design. Authors simply mention that sampling will be probability proportional to size and influenced by urbanization. What is the actual sampling methodology employed and how does this give the authors confidence that they will be able to construct a representative sample?
- The authors mention that data will be collected through telephone interviews (line 302). This invites the selection bias of excluding people who do not have access to a telephone. This would be a significant limitation to the data because of how vulnerable populations are disproportionately impacted by COVID. Can the authors please comment on why they are not making allowances for people without telephone access and the magnitude of the bias that this will invite into the study?
- What will the authors cluster on for the random effects analysis (line 476-477)?
- Why are the authors using random effects models instead of generalized estimating equations?
- Section 2.13 Scoping Review is separate from the rest of the paper, which describes the data set that is being constructed. Authors should consider omitting this from the manuscript.
Author Response
Dear Reviewer,
We would like to thank you for your interest in our article and for the time that you have devoted to reviewing it.
We are pleased to submit a revised version of our manuscript. We have highlighted the changes by using the track changes mode in MS Word.
Below we respond to your comments:
COMMENT: As a global comment, the authors should be mindful of runoff sentences. Many of the sentences in the manuscript were too long, with multiple ideas communicated, making it difficult to maintain focus. Please see below for specific comments.
RESPONSE: Thank you for your suggestion. We have shortened long sentences and highlighted the changes by using the track changes mode in MS Word. However, because of time constraints there will still be some long sentences as this is a style issue that would require more time than we have to resolve.
INTRODUCTION
RESPONSE: Following the comments of the rest of the reviewers, the background has been reduced by eliminating some of the sentences. The information requested has been incorporated below in response to your comments. We hope this is acceptable, although, if you consider it essential, we could also include the answers in the article itself.
COMMENT: What are the lifestyle habits that contributed to further stigmatization of the Roma population? (line 69)
RESPONSE: The life events/lifestyle habits referred to are, for example, the massive number of celebrations such as engagement parties, weddings, funerals etc., as well as the street vending activities which make up the main occupation of this population. Further details can be found in reference 20.
COMMENT: What is the vulnerable population that is referenced in the last sentence of the second paragraph (lines 71-72)?
RESPONSE: This paragraph has been modified in the text, indicating that the vulnerable groups to which we refer are minors in foster care, ethnic minorities, the elderly or people with chronic diseases (lines 64-65).
COMMENT: Are the populations highlighted in the second paragraph (Roma, infants and adolescents, other vulnerable) all in Spain or is there a different/broader geographic distribution?
RESPONSE: In the specific case of this paragraph, the scope of the references is Spain, although in the following sections, where the information for each group is expanded, international studies have also been included.
COMMENT: Why are they authors highlighting these specific vulnerable populations? What about these populations make them vulnerable (in general and in the context of COVID)?
COMMENT: Authors should provide more detail (statistics) for how COVID and the lockdowns are already known to have impacted people with chronic conditions.
RESPONSE: People with chronic illnesses account for about 50 percent of all GP visits, 64 percent of all outpatient visits, and more than 70 percent of all hospital days, so if you restrict medical assistance, this population group is directly affected. For instance, some examples of statistics for people with chronic diseases: 31% with epilepsy indicated that the frequency of seizures increased during confinement (ref 27); 30% of Alzheimer's patients and 40% of their caregivers reported a deterioration in the health status of the patients during confinement (ref 29).
COMMENT: Considering the limitations of non-probabilistic samples and web surveys described by the authors, what are the substantive scientific benefits to using such modes of data collection (beside the speed at which data can be collected and reported)?
RESPONSE: These modes of data collection have the advantages of having a much larger than usual sample size and being less costly. In addition, online surveys offer a substantial decrease in the time needed to achieve a given sample size (Ilieva et al. 2002). Another advantage is the computerization of the questionnaire, which entails a wide spectrum of possibilities in terms of flux control, multimedia content. etc. A final advantage of web surveys is the fact that they may be helpful to find members of non-demographic strata in a population and can be used to find respondents from read-to-reach populations. Some people have even come to believe that probability surveys could gradually disappear, but probability surveys continue to be fundamental for producing official statistics in the current context of surveys carried out by national statistical agencies, where high data collection costs and increasingly lower response rates are observed, since web alternatives are not the same in reliability and general enough to eliminate the use of probability surveys without severely sacrificing the quality of the estimates (Beaumont,2020).
Beaumont, J.-F. Are probability surveys bound to disappear for the production of official statistics?. Survey Methodology, Statistics Canada. 2020; Catalogue No. 12-001-X, Vol. 46, No. 1. Available at: http://www.statcan.gc.ca/pub/12-001-x/2020001/article/00001-eng.htm.
Ilieva, J., Baron, S. Healey, NM. Online surveys in marketing research. International Journal of Market Research. 2002; 44 (3), 1-14.
COMMENT: Can the authors please explain what they mean by “transversal designs”? (line 96)
RESPONSE: Thank you. This was a mistake in the English translation. We have now changed it to ‘cross-sectional designs’ (line 106).
COMMENT: What exactly is meant by “continuous information on the evolution of the epidemic” (line 110-111)?
RESPONSE: Thank you. We changed it as “monitoring the evolution of the epidemic”. (line 104)
COMMENT: In line 119, authors should qualify what they mean by “greatest impact”. In previous paragraphs, authors gave examples for how COVID impacted other populations (in different ways). With the over 70 years of age group, are they referring to mortality?
What is the reason for the sex disparity in COVID-19 mortality (lines122-123)?
RESPONSE: This sentence has been redrafted in the text of the article for better understanding. This population has been the one with the greatest representation, whether we talk about incidence, hospitalization and/or death from COVID in Spain. According to the article referred to, the higher mortality in men (than in women) could be due to more comorbidities and risk factors (e.g., smoking, obesity), and also to differences in cellular immunity between men and women, including a poorer T-cell activation and increased pro-inflammatory cytokines in men (ref 36)
COMMENT: What is the “specific, reliable and timely” data that is being provided by the Health Care and Social Survey? (lines 140-141)
RESPONSE: Thank you. We have erased those terms from that sentence (line 120).
COMMENT: What do the authors mean when they say, “…it is populated as other European countries such as Austria or Switzerland.”? (line172-173)
RESPONSE: We noted that Andalusia was an outstanding region of Europe in terms of population. We have rewritten that sentence and erased the reference to those countries.
COMMENT: Section 1.3 reads as statements of fact. Authors should re-write this sentence to clarify that these are all testable hypotheses that the data set will answer. Also, these do not seem like hypotheses because some of this was discussed earlier in the introduction as findings from recent research.
RESPONSE: Thank you. This has now been redrafted in terms of hypotheses to be tested.
COMMENT: (line 202) What is the timeframe for short and mid-term impact?
RESPONSE: By short- and mid-term we mean to one year and three years, respectively. We have included that timeframe the first time these terms appear in the test (line 161).
COMMENT: What do the authors mean by psychosocial and behavioral determinants of health (as these can also be health outcomes)? (line 204)
RESPONSE: Effectively, they can also be health outcomes. We specified some of them in Table 5 of information blocks and variables. To identify them, we followed the reference [50] “Cabrera-León A, Daponte Codina A, Mateo I, Arroyo-Borrell E, Bartoll X, Bravo MJ, et al. Contextual indicators to assess social determinants of health and the Spanish economic recession. Gac Sanit. 2017 May 1;31(3):194–203” included in lines 124 and 562.
METHODS
COMMENT: Can the authors provide more background for what is considered a disadvantaged area (line 225)?
RESPONSE: We have now included the sentence, "Disadvantaged areas are defined according to criteria of social (unemployment, immigration or social disintegration), urban planning (housing) and educational (illiteracy, absenteeism or school failure)." in lines 243-245, and we cite reference [50] for further information.
COMMENT: The omission of nursing homes will be a significant limitation to this data, considering the impact of COVID on the elderly (line 227). Do the authors plan to just accept this limitation or is there a way for them to augment the data set with this information to understand the impacts of COVID?
RESPONSE: We totally agree with you. This study was initially designed to also include the population from nursing homes, but this was ultimately not possible. This is an unfortunate limitation of the study considering the enormous impact that COVID-19 has had on this population. We have added this limitation in lines 591-563.
COMMENT: Why did the authors skip 24 months as a timepoint for data collection? (line 233)
RESPONSE: We skipped the same 24 months in the populations studied (ESSOCgeneral, zones and +55) to gather information in the mid-term from the initiation of the state of alarm in Spain started, as well as to compare their results in the same timepoint. We have added that in lines 227-229.
COMMENT: Why do the residents of the ESSOCzones only have two points of data collection compared to ESSOCgeneral? Considering the presumed vulnerability of the residents in the ESSOCzones, it would be important to collect just as much information on these people as for the ESSOCgeneral. Same comment applies to the ESSOC+55 group.
RESPONSE: The first measurements are different for each ESSOC survey depending on the availability of the sampling frames. Therefore, ESSOCgeneral was the first survey to be carried out because the sample frame was already available, while the ESSOC+55 sampling frame had to be developed from scratch and the ESSOCzones sampling frame had to be integrated into the same ESSOC general sampling frame in order to geographically identify the population located in disadvantage zones. We explain this in lines 236-242. On the other hand, the sampling frame of disadvantaged areas was made available at the end of measurement 4 of the ESSOCgeneral, thus we have been able to apply post-stratification according to whether or not the selected person lives in those areas. In this way, more representative results of the population residing in disadvantaged areas that were selected for the ESSOCgeneral are provided. Furthermore, in the intermediate analyzes that we are currently carrying out, we will check to what extent we can even offer precise estimates of this population living in disadvantaged areas. If this is the case, we will then be able to provide results for that population from the first measurement of the general ESSOC as well. We are currently carrying out these analyzes.
COMMENT: Why do the authors expect such a high non-response rate (40%)? (lines 261-262). If the authors are anticipating so much non-response and loss to follow-up, can they modify the study design to improve participation?
RESPONSE: Sample size for the first measurement of the ESSOCgeneral was estimated according to a non-response rate of 40%. This response rate was considered as the percentage of surveys carried out or effective sample with respect to the total of the theoretical sample. Thus, this calculation includes in the denominator all types of incidents, that is, not only those directly related to the refusal of the selected person to be interviewed, but also those incidents:
- from the sampling frame (e.g., incorrect or unreachable telephone number, or person residing in a place other than the one selected);
- when conducting the interview (e.g., person who cannot be reached, appointment outside the fieldwork time, or insufficient number of attempts to finish the fieldwork, that is, most would be cases that would have answered the interview if it had been extended the end of the fieldwork);
- leaving the cohort (e.g., deceased, long-term hospital admissions, change of residence etc).
The reason for including all types of incidents in the response rate is to obtain a sufficient theoretical sample size to reach the effective sample size of 3,000 interviews. This means that, once almost all the fieldwork of the longitudinal sample had progressed, a new sample was added until the total effective size of 3,000 interviews was reached. When that sample size was achieved, the field work was completed at each measurement.
In short, if we take into account only the refusals of the selected person, our total response rates are above 70%. Moreover, the field work carried out has introduced elements of improvement that have allowed for an increase in the response rates from one measurement to the next, both for the longitudinal samples and for the new ones that were incorporated in each measurement.
Thus, for measure 2 of the general ESSOC, the response rate for the longitudinal sample was 78.3% and for the new one, 58.2% (total 73.0%). For measure 3 it was 88.8% and 75.3%, respectively (total 83.1%), and for measurement 4 it was 93.1% and 82.9% (total 90.6%). To clarify this, we have included these comments in the article (lines 291-319) and have corrected Figure 1 with the response rates calculated only with the negatives, noting this in the figure itself.
COMMENT: How do the authors plan to account for the mixed samples at each sampling event (one group is maintained from the prior sampling, the other group is newly sampled)?
RESPONSE: We are not sure we fully understand your question. We believe the answer would be in sections 2.4 Sample Size and 2.5 Sample selection and sample allocation which have been improved thanks to your following comments.
COMMENT: “2.5 Sample Allocation” more details need to be provided regarding the sampling design. Authors simply mention that sampling will be probability proportional to size and influenced by urbanization. What is the actual sampling methodology employed and how does this give the authors confidence that they will be able to construct a representative sample?
RESPONSE: All the samples are selected by random stratified sampling and the allocation is carried out in such a way that within each stratum the samples are self-weighted. The calculations of the estimators are carried out considering the design weights, so that the estimates of the totals and proportions, as well as the associated sampling errors, are unbiased from the respective parameters. Calibration is also applied to obtain more reliable estimates based on the demographic characteristics of the study population (Andalusia, Spain).
Sections 2.5 and 2.6 were rewritten and now include more information about the probability sampling design. In addition, both sections (Sample Allocation and Sample Selection) were merged to make them clearer (lines 321-347).
COMMENT: The authors mention that data will be collected through telephone interviews (line 302). This invites the selection bias of excluding people who do not have access to a telephone. This would be a significant limitation to the data because of how vulnerable populations are disproportionately impacted by COVID. Can the authors please comment on why they are not making allowances for people without telephone access and the magnitude of the bias that this will invite into the study?
RESPONSE: We totally agree with you. In fact, we included that limitation in the Discussion section (lines 575-580) as follows:
“…The limitations of the study include those derived from the coverage and quality of the sampling frames, which may cause selection biases. In the case of the BDLPA, its telephone coverage is over 90% and it tends to have low percentages of non-existent telephone numbers (7%–8%) or non-contactable telephone numbers due to the frame’s outdatedness (9%–10%). That said, any potential biases due to such defects will be corrected through the estimator calibration techniques described above. Another limitation is that caused…”
In addition, other ways of gathering data, such as face-to-face interviews, were not possible due to confinement restrictions in place because of the COVID-19 pandemic. Moreover, surveys based on face-to-face interviews also have a lack of coverage of people living in highly disadvantaged areas, so such population-based surveys also have difficulties in collecting data from extremely vulnerable populations.
COMMENT: What will the authors cluster on for the random effects analysis (line 476-477)?
RESPONSE: We have explained this point better in the new version of the manuscript (lines 525-528).
(…) when the variable is discrete (equivalent to a Poisson regression). Random effects will be included in these models to capture the effects of unobserved confounders. Random effects will be included in these models with two goals: first, capture the effects of unobserved confounders, and second, capture the time dependency (as you will get repeated measures of the subjects) and the spatial dependency (i.e., spatial clustering), if applicable.
Inferences will (…)’
COMMENT: Why are the authors using random effects models instead of generalized estimating equations?
RESPONSE: Generalized estimating equations is a marginal approach that does not allow inferences at the individual level, whereas random effects is a conditional approach that, allows inferences to be made at the individual level (as well as at the 'population' level as the methods of the marginal approach). Some of the estimators (of time dependence and spatial clustering, for example) interest us at different levels, including the individual. Furthermore, the marginal approach is less efficient than the conditional approach.
COMMENT: Section 2.13 Scoping Review is separate from the rest of the paper, which describes the data set that is being constructed. Authors should consider omitting this from the manuscript.
RESPONSE: Following your suggestion, we decided not to include the last objective and its related section in the manuscript.
Once again, thank you for reviewing our article and for your comments, which have undoubtedly improved the quality of the manuscript. We hope we have addressed the shortcomings that you identified.

Reviewer 3 Report
The manuscript "Impact COVID-19 on the health of the general and more vulnerable population and its determinants: Health care and social survey-ESSOC, study protocol” describes the rationale and protocol of a population study aiming at determine the COVID-19 impact on overall health as well as the socioeconomic, psychosocial, occupational , environmental and clinical determinants of both the general and more vulnerable population.
The question posed by the authors is well defined and the article appears to fit the aims and scope of the journal. The description of the protocol design is clear and exhaustive.
I think that the article is suitable for publication.
I would just suggest that the authors summarize the background section by citing previous experiences but without describing them in detail. In the present form the introduction appears quite extensive while it should be focused above all on the ESSOC study framework.
Please let a native speaker to improve English language.
Author Response
Dear Reviewer,
We would like to thank you for your interest in our article and for the time that you have devoted to reviewing it.
We are pleased to submit a revised version of our manuscript. We have highlighted the changes by using the track changes mode in MS Word.
Below we respond to your comments:
The manuscript "Impact COVID-19 on the health of the general and more vulnerable population and its determinants: Health care and social survey-ESSOC, study protocol” describes the rationale and protocol of a population study aiming at determine the COVID-19 impact on overall health as well as the socioeconomic, psychosocial, occupational, environmental and clinical determinants of both the general and more vulnerable population.
The question posed by the authors is well defined and the article appears to fit the aims and scope of the journal. The description of the protocol design is clear and exhaustive.
I think that the article is suitable for publication.
COMMENT: I would just suggest that the authors summarize the background section by citing previous experiences but without describing them in detail. In the present form the introduction appears quite extensive while it should be focused above all on the ESSOC study framework.
RESPONSE: Following your suggestion, the text from the studies found has been reduced by only referring to them rather than specifying too many details about the findings.
COMMENT: Please let a native speaker to improve English language.
The manuscript has been reviewed by a native speaker, who has corrected errors and endeavored to maintain the original style of the authors.
Once again, thank you for reviewing our article and for your comments, which have undoubtedly improved the quality of the manuscript. We hope we have addressed the shortcomings you identified.
